



# Robustness of Neural Network Emulations of Radiative Transfer Parameterizations in a State-of-the-Art General Circulation Model

Alexei Belochitski[1,2], Vladimir Krasnopolsky[2]

[1]IMSG, Rockville, MD 20852, USA

[2]NOAA/NWS/NCEP/EMC, College Park, MD 20740, USA

*Correspondence to*: Alexei Belochitski (alexei.a.belochitski@noaa.gov)

**Abstract.** The ability of Machine-Learning (ML) based model components to generalize to the previously unseen inputs, and the resulting stability of the models that use these components, has been receiving a lot of recent attention, especially when it comes to ML-based parameterizations. At the same time, ML-based emulators of existing parameterizations can be stable,

accurate, and fast when used in the model they were specifically designed for. In this work we show that shallow-neural-network-based emulators of radiative transfer parameterizations developed almost a decade ago for a state-of-the-art GCM are robust with respect to the substantial structural and parametric change in the host model: when used in two seven month-long experiments with the new model, they not only remain stable, but generate realistic output. Aspects of neural network architecture and training set design potentially contributing to stability of ML-based model components are discussed.

## 1 Introduction

One of the main difficulties in developing and implementing high-resolution environmental models is complexity of the physical processes involved. For example, the calculation of radiative transfer in a GCM often takes a significant part of the total model run time. From the standpoint of basic physics, radiative transfer is well understood. Very accurate, but computationally complex benchmark models exist (Oreopoulos et al, 2012) that demonstrate excellent agreement with

observations (Turner et al, 2004). Parameterizations of radiative transfer seek a compromise between accuracy and computational performance. Arguably, the biggest simplification they make is treatment of radiative transfer as a 1-D as opposed to a 3-D process (Independent Column Approximation, ICA): both solar, or short-wave (SW), and terrestrial, or long-wave (LW), radiation is considered to flow within the local column of the model, up and down the local vertical (two stream approximation), but not between columns. This approximation works well at spatial resolutions characteristic of general

circulation models of the atmosphere (Marshak and Davis, 2005). To integrate over the spectrum of radiation, parameterizations split it into several broad bands and a number of representative spectral intervals that are treated monochromatically (Fu and Liou, 1992). State-of-the-art parameterizations can reproduce benchmark calculations to a high degree of accuracy even with these simplifications, but they still require substantial computational expense.



Radiative transfer parameterizations supply their host model with broadband fluxes and heating rates, which are obtained by
integration over time, space, and frequency. Therefore, a trade-off between accuracy and computational expense can be found
in how finely these dimensions are discretized (Hogan et al, 2017).

- *Discretization in time.* All GCMs update their radiative heating/cooling rates less frequently than the rest of the model
  fields. For example, National Centers for Environmental Prediction (NCEP) Global Forecast System (GFS) v16
  general circulation model (GCM) in its operational configuration updates its radiative fields once per model hour,
while updates to temperature, moisture, most cloud properties etc. due to unresolved physics processes happen every
  150 model seconds, or 24 times per single radiation call. Updates due to dynamical processes happen even more
  frequently, every 12.5 seconds (Kain et al, 2020). This approximation is good for slowly changing fields of certain
  radiatively active gases but is less justified for small-scale clouds with lifetimes of an hour or less.

- *Discretization in space.* Some GCMs calculate radiative fields on a coarser spatial grid and interpolate them onto a
finer grid used for the rest of the model variables. For example, radiation grid in European Centre for Medium-Range
  Weather Forecasts (ECMWF) Integrated Forecast System (IFS) v43R3 in the ensemble mode is 6.25 times coarser
  than the physics grid (Hogan et al, 2017). This may cause 2m temperature errors in areas of surface heterogeneity,
  e.g. coasts (Hogan and Bozzo, 2015).

- *Discretization/sampling in frequency space.* The Rapid Radiative Transfer Model (RRTMG), a parameterization of
radiative transfer for GCMs used in NCEP GFS and ECMWF IFS, utilizes 14 bands in the short wave (Mlawer et al,
  1997), while the parameterization used at United Kingdom Met Office Unified Model utilizes 6 (Edwards and Slingo,
  1996). Monte Carlo Spectral Integration (Pincus and Stevens, 2009) performs integration over only a part of the
  radiative spectrum, randomly chosen in each point in time and space, allowing to increase temporal/spatial resolution
  of radiation calculations. Monte Carlo Integration of the Independent Column Approximation (McICA) (Pincus et al,
2003), integrates over the entire spectrum, but samples subgrid-scale (SGS) cloud properties in a random, unbiased
  manner, in each grid column in time and space, instead of integrating over them.

All of the methods for improving computational efficiency of radiative transfer parameterizations outlined above are either
numerical and/or statistical in nature. In recent years there has been a substantial increase in interest in adding machine learning
(ML) techniques to the arsenal of these methods. It has been accomplished in at least two different ways: 1) as an emulation
technique for accelerating calculations of radiative transfer parametrizations or their components, 2) as a tool for development
of new parameterizations based on data simulated by more sophisticated models and/or reanalysis.

An ML-based emulator of a model physics parameterization is a functional imitation of this parameterization in a sense that
the results of model calculations with the original parameterization and with its ML emulator are physically identical. It is
accomplished by using the data simulated by running the original model with the original parameterization for ML emulator





training, which allows to achieve a very high accuracy of approximation because simulated data are free of the problems typical of empirical data.

Unbiased, random, uncorrelated errors in radiative heating rates do not statistically affect forecast skill of an atmospheric model (Pincus et al, 2003). From the physical standpoint this can be understood in the following way: random small local heating rate errors in the bulk of the atmosphere lead to local small-scale instabilities that are mixed away by the flow; however, there no such mechanism for the surface variables, such as skin temperature, and errors in surface fluxes can be more consequential (Pincus and Stevens, 2013). Therefore, it may be useful to think of the above as necessary conditions on

approximation error of an ML emulator for it to be a successful functional imitation of a parameterization.

NeuroFlux, a shallow-neural-network-based LW radiative transfer parameterization, developed at ECWMF, was in part an emulator and in part a new ML-based parameterization (Chevallier et al, 1998, 2000). It consisted of multiple neural networks (NNs), each utilizing a hyperbolic tangent as an activation function (AF), but using a varying number of neurons in the single

hidden layer: two NNs were used to generate vertical profiles of, respectively, up- and down-welling clear sky LW fluxes; and a battery of NNs, two per each vertical layer of the host model, was used to compute profiles of up- and down-welling fluxes due to blackbody cloud on a given layer, with overall fluxes calculated using the multilayer graybody model. Training set for clear-sky NNs contained 6000 cloudless profiles from global ECMWF short-range forecasts; one day worth of three-hourly data per month of a single year was utilized. From this set, multiple training sets for cloudy sky NNs were derived, each

containing 6000 profiles as well: a blackbody cloud was artificially introduced on a given vertical layer, and radiative transfer parametrization was used in the offline mode to calculate resulting radiative fields. NeuroFlux was accurate and about an order of magnitude as fast as the original parameterization in a model with 31 vertical layers. It had been used operationally within the ECMWF four-dimensional variational data assimilation system (Janiskova et al, 2002). However, in model configurations with 60 vertical layers and above, NeuroFlux could not maintain the balance between speed up and accuracy (Morcrette et al,

85 2008).

The approach based on pure emulation of existing LW- and SW-radiative transfer parametrizations using NNs has been pursued at NCEP Environmental Modeling Center (Krasnopolsky et al, 2008,2010, 2012; Belochitski et al, 2011). In this approach, two shallow NNs with hyperbolic tangent activation functions, one for LW and the other for SW radiative transfer,

generate heating rate profiles as well as surface and top-of-the-atmosphere radiative fluxes, replacing the entirety of respective RRTMG LW and SW parameterizations. Not only radiative transfer solvers were emulated, but also the calculations of gas and cloud optical properties (aerosol optical properties were prescribed from climatology). Two different pairs of emulators were designed for two different applications: climate simulation and medium-range weather forecast, each differing in the training set design. The data base for the former application was generated by running NCEP Climate Forecast System (CFS),

a state-of-the-art fully coupled climate model, for 17 years (1990-2006) and saving instantaneous inputs and outputs of





RTTMG every three hours for one day on the 1st and the 15th of each month, to sample diurnal and annual cycles, as well as decadal variability and states introduced by time-varying greenhouse gases and aerosols. 300 global snapshots were randomly chosen from this database, and consequently split into three independent sets for training, testing, and validation, each containing about 200,000 input/output records (Krasnopolsky et al, 2010). The data set for the medium range forecast

application was obtained from 24 10-day NCEP GFS forecasts initialized on the 1st and the 15th of each month of 2010, with each forecast saving instantaneous three-hourly data. Independent data sets were obtained following the same procedure as for the climate application (Krasnopolsky et al, 2012).

Dimensionality of data sets and NN input vectors for both applications was reduced in the following manner: some input

profiles (e.g. pressure) that are highly correlated in the vertical were sampled on every other level without decrease in approximation accuracy; some inputs that are uniformly constant above certain level (water vapor) or below a certain level (ozone) were excluded from the training set on these levels; inputs that are given by prescribed monthly climatological look up tables (e.g. trace gases, tropospheric aerosols) were replaced by latitude and periodic functions of longitude and month number; inputs given by prescribed monthly time series (e.g. carbon dioxide, stratospheric aerosols) were replaced by the year

number and periodic function of month number. No reduction of dimensionality was applied to outputs.

A very high accuracy and up to two orders of magnitude increase in speed as compared to the original parameterization for both NCEP CFS and GFS full radiation has been achieved for model configurations with 64 vertical levels. The systematic errors introduced by NN emulations of full model radiation were negligible and did not accumulate during the decadal model

simulation. The random errors of NN emulations were also small. Almost identical results have been obtained for the parallel multi-decadal climate runs of the models using the NN and the original parameterization, and in the limited testing in the medium-range forecasting mode. Regression trees were explored as an alternative to NNs and were found to be nearly as accurate in a 10-years long climate run while requiring much more computer memory due to the fact that the entire training data set has to be stored in memory during model integration (Belochitski et al, 2011).


Using the approach developed at NCEP, an emulator of RRTMG consisting of a single shallow NN that replaces both LW and SW parameterizations at once was developed at Korean Meteorological Agency for the short-range weather forecast model Korea Local Analysis and Prediction System in an idealized configuration with 39 vertical layers (Roh and Song, 2020). Inputs and outputs to RRTMG were saved on each 3 second time step of a 6 hour-long simulation of a squall line, and about 270,000

input/output pairs were randomly chosen from this data set to create training, validation, and testing sets. Dimensionality reduction was performed by removing constant inputs. Several activation functions were tested (tanh, sigmoid, softsign, arctan, linear) with hyperbolic tangent providing best overall accuracy of approximation. The emulator was two orders of magnitude as fast as the original parameterization, and was stable in a 6 hour-long simulation.



Two dense, fully-connected, feed-forward deep-NN-based emulators with three hidden layers, one emulator per parametrization, were developed for LW and SW components of RRTMG-P for the Department of Energy's Super-Parametrized Energy Exascale Earth System Model (SP-E3SM) (Pal et al, 2019). In SP-E3SM, radiative transfer parameterizations act in individual columns of a 2-D cloud resolving model with 31 vertical levels embedded into columns of the host GCM. Calculation of cloud and aerosol optical properties were not emulated, instead, original RRTMG-P subroutines were used. Inputs and outputs of radiative parameterizations were saved on every time step of a year-long model run, with 9% of this data randomly chosen to form a data set of 12,000,000 input/output records for LW, and of 6,000,000 input/output records for SW emulator training and validation. 90% of the data in these sets was used for training, and 10% for validation and testing. No additional dimensionality reduction was performed. Sigmoid AF was chosen as it was found to provide slightly better training convergence than the hyperbolic tangent. The emulator was an order of magnitude faster than the original parametrization and was stable in a year-long run.

Recently, an entire suite of model physics in NCEP GFS, that among other parameterizations includes LW and SW RRTMG schemes, was emulated with a single shallow NN. It was shown that the emulator is accurate and stable in multiple 10-day forecasts and in one-year continuous run (Belochitski and Krasnopolsky, 2021).

A number of ML-based radiative transfer parameterizations or their components have been developed, but, to our knowledge, have not yet been tested in an online setting, or in interactive coupling to an atmospheric model. Among them are deep-NN-based parameterizations of gas optical properties for RTTMG-P (Ukkonen et al, 2020; Veerman et al, 2021), and a SW radiative transfer parameterization based on convolutional deep neural networks (Lagerquist et al, 2021).

The domain of the mapping approximated by an NN is defined not only by the parameterization that is being emulated, but by the entirety of the atmospheric model environment: the dynamical core, the suit of physical parameterizations, and the set of configuration parameters for both. Once any of these components is modified, the set of possible model states is modified as well, possibly now including states that were absent in the NN's training data set.

In this work we evaluate robustness of ML emulators of radiative parameterizations. We investigate how much of a change in the model's phase space (and NN domain) a statistical model like the NN can tolerate. We will approach this question by installing shallow-NN-based emulators of LW and SW RRTMG developed in 2011 for NCEP CFS (Krasnopolsky et al, 2010) into the new version 16 of NCEP GFS that became operational in March of 2021. In Section 2 we outline the differences between the 2011 version of CFS and the new version of GFS. In Section 3, experiments performed with SW and LW emulators developed for the 2011 version of CFS and incorporated into GFS v16 are described and their results are examined. Section 4 discusses aspects of neural network architecture and training set design potentially contributing to stability of ML-based model components. Conclusions are formulated in Section 5.



## 2. Differences between the atmospheric component of 2011 NCEP CFS and 2021 version of NCEP GFS

GFS v16 differs from the 2011 version of the atmospheric component of NCEP CFS in a number of ways, most relevant of which are summarized in Table 1.

|  | CFS 2011 | GFS 2021 |
|---|---|---|
| Dynamical core | Spectral Eulerian | Finite Volume Cubed Sphere |
| Horizontal resolution | T126 (~100 km) | C768 (~13 km) |
| Vertical res. and coordinate | 64 levels, hybrid sigma-p | 127 levels, hybrid sigma-p |
| Physics Grid | Gaussian | Cubed Sphere |
| Radiation | RRTMG v2.3 | RRTMG LW v4.82, SW v3.8 |
| Microphysics | Zhao-Carr, single moment, two species, one prognostic variable | GFDL, single moment, five species, five prognostic variables |
| Planetary Boundary Layer | K-profile | Hybrid TKE-EDMF |
| Middle atm. $H_2O$ photochemistry | None | Climatological |
| $O_3$ photochemistry | None | Climatological |
| Stratospheric aerosols | Time-dependent, prescribed | |
| Tropospheric aerosols | Climatological | |
| $CO_2$ | Time-dependent, prescribed | |
| Trace gases | Climatological | |

**Table 1. Differences between the atmospheric component of 2011 NCEP CFS and 2021 version of NCEP GFS**


From the standpoint of implementation of radiative transfer emulators developed in 2011 into the modern generation of GFS, the most consequential change in the model is the near doubling of the number of vertical layers because it has a direct impact on the size of the input layer of the NN-based emulator. Therefore, we reconfigure GFS v16 to run with 64 layers in the vertical.

Another consequential change in the model appears to be replacement of the Zhao-Carr microphysics (Zhao and Carr, 1993) with the GFDL scheme (Lin et al, 1983; Chen and Lin, 2011; Zhou et al. 2019). Using the latter in combination with 2011 RTTMG emulators resulted in unphysical values of outgoing LW radiation at the top of the atmosphere (TOA) (not shown). Potential explanation is that the change in microphysical parameterization leads to an increase in the number of the model's prognostic variables. Both the spectral and the finite volume dynamical cores include zonal and meridional wind components,


pressure, temperature, water vapor and ozone mixing ratios as prognostic variables. Zhao-Carr microphysics adds only one

more prognostic to this list: mixing ratio of total cloud condensate (defined as the sum of cloud water and cloud ice mixing

ratios). GFDL microphysics adds 6 prognostic variables: cloud water, cloud ice, rain, snow, and graupel mixing ratios, as well

as cloud fraction. The near doubling of the number of prognostic variables from 7 to 12 leads to the proportional increase in

dimensionality of the physical phase space of the model. As a result, the set of possible model states in GFS v16 is very

different, from a mathematical standpoint, than in the 2011 CFS. Even though the vector of inputs to the LW parameterization

remains the same in the new model, it is obtained by mapping from a very different mathematical object, potentially increasing

the probability that a given input vector lies outside of the NN's original training data set domain. For our experiments, we

replaced the GFDL microphysical parametrization with the Zhao-Carr scheme.

The new hybrid TKE-EDMF planetary boundary layer (PBL) parameterization (Han and Bretherton, 2019) also introduces a

new prognostic variable, sub-grid scale turbulent kinetic energy, that was absent in the 2011 version of CFS. Even though we

did not see adverse effects stemming from the use of the new PBL scheme in preliminary testing, we replaced it with the

original K-profile/EDMF scheme (Han and Pan, 2011) out of abundance of caution.

$CO_2$ concentration values used during training of 2011 emulators ranged from 350 to 380 ppmv between the years 1990 and

2006, respectively. In our current experiments spanning 2018, $CO_2$ concentration was about 409 ppmv.

There were incremental updates and parametric changes to all other components of the suite of physical parameterizations, too

numerous to be listed here; in addition, model's software infrastructure was completely overhauled, including the new ESMF-

based modeling framework, coupler of the dynamical core to the physics package, input/output system, and workflow scripts

(see the document GFS/GDAS Changes Since 1991, https://www.emc.ncep.noaa.gov/gmb/STATS/html/model_changes.html)

## 3. Results

In addition to changes described in the previous section (reconfiguring the model to use 64 vertical layers, replacing GFDL

microphysics with the Zhao-Carr scheme, and hybrid TKE-EDMF parameterization with a K-profile/EDMF PBL scheme), we

configure GFS v16 to run at C96 horizontal resolution (~100km) to reduce computational expense of the model. This

configuration will be referred to GFS in the following discussion and was used in control runs. We then replaced both modern

versions of LW and SW RTTMG parametrizations in GFS with radiative transfer emulators developed in 2011. This version

of the model will be referred to as hybrid deterministic-statistical GFS, or HGFS. Two 7-month long runs were performed with

each model configuration: one initialized on 1/1/2018 and the other one on 7/1/2018, both using 2018 values of radiative

forcings, with the instantaneous output saved 3-hourly. Sea surface temperatures (SSTs) in GFS forecasts are initialized from

analysis and relax to climatology with 90-day e-folding time scale as forecast progresses. First 30 days of each of the two 7



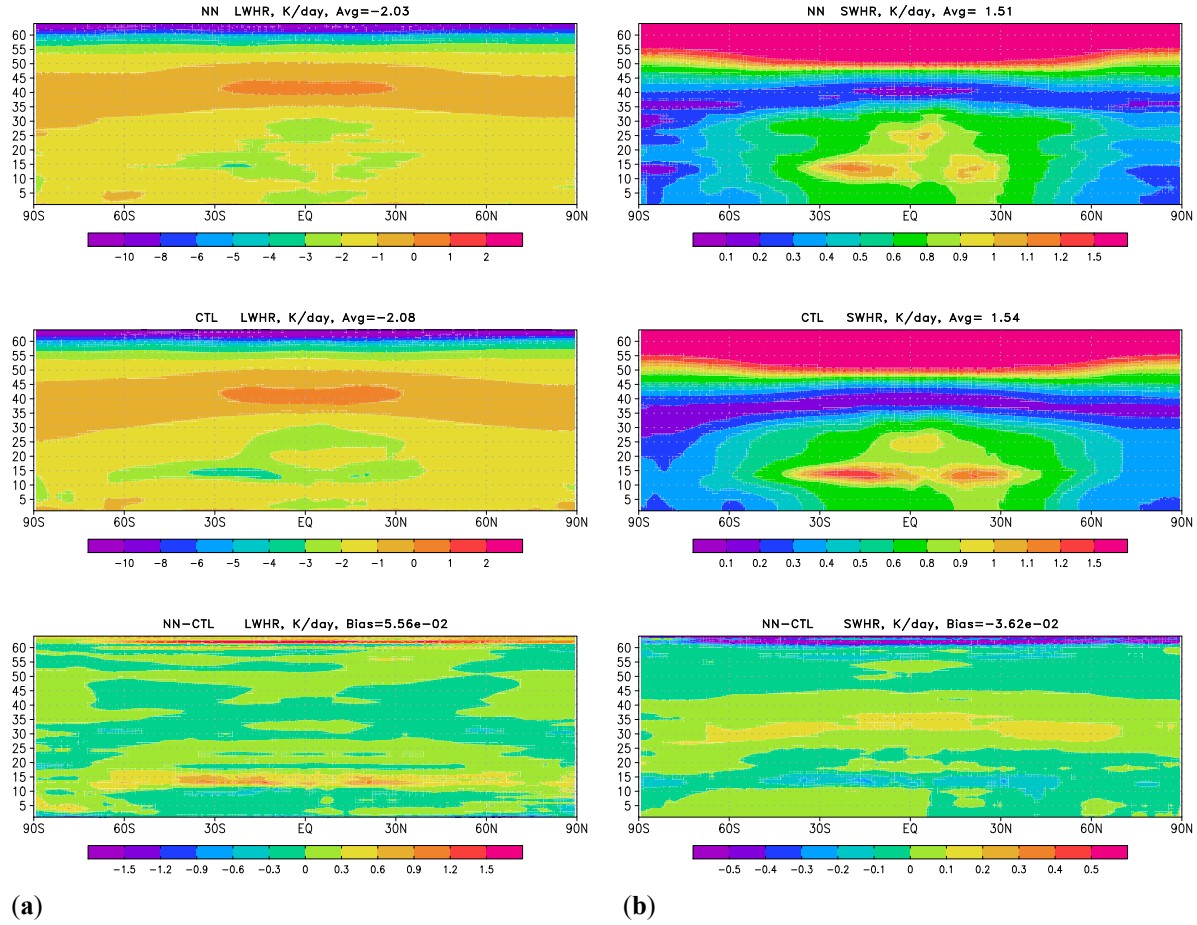

**Figure 1. Zonal and time mean over 12-month AMIP-like run covering 2018 for: (a) Long-wave heating rate, K/day; (b) Short-wave heating rate, K/day. Upper row – results produced by HGFS, medium – by GFS, and the lower row the difference (HGFS – GFS). Vertical coordinate shows model level number.**

month-long runs were discarded, and remaining 6 months of data in each experiment were combined into a single data set mimicking a 12 month-long run forced by climatological SSTs. Note, that the 7-month length of each run was determined exclusively by the 8-hour wall clock time limit for a single run on the NOAA RDHPCS Hera supercomputer used for our experiments.

Figure 1 shows zonal and time mean over 12 months of AMIP-like run, covering the year of 2018 for LW (left panel) and SW (right panel) heating rates. Global biases are small for both heating rates and constitute about 2-3% of the global mean value. Decrease in LW radiative cooling at the top of tropical and subtropical boundary layer is compensated by the corresponding decrease in SW radiative heating, and consistent with decrease in low cloud cover in these areas (not shown). Biases in the stratopause may be related to the new parameterizations of O3 and H2O photochemistry that were not present in the 2011

version of the model.



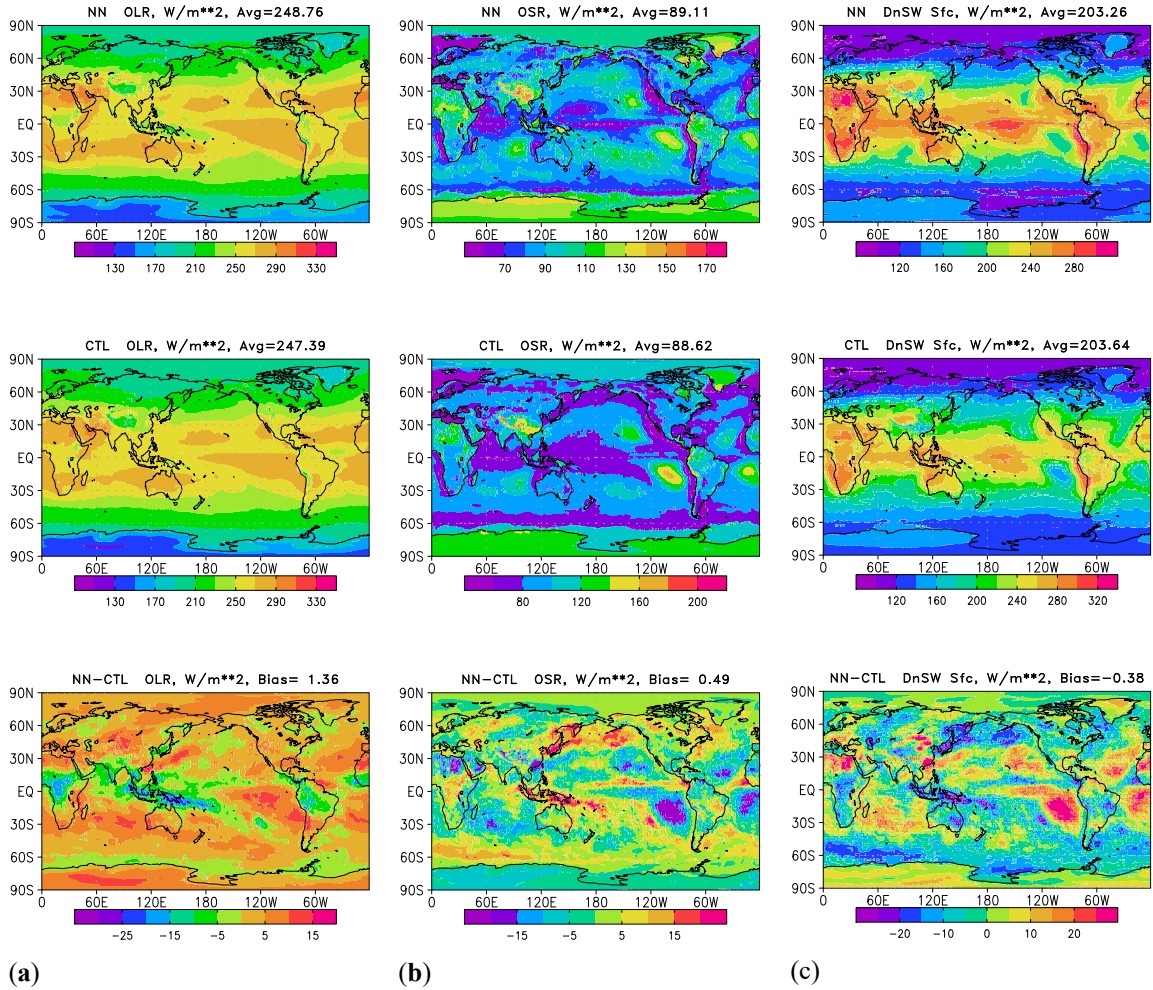

**Figure 2. Zonal and time mean over 12-month AMIP-like run covering 2018 for: (a) Outgoing LW radiation at the TOA; (b) Outgoing SW radiation at the TOA; (c) Downwelling SW radiation at the surface. Upper row – results produced by HGFS, medium – by GFS, and the lower row the difference (HGFS – GFS). Vertical coordinate shows model level number.**

Figure 2 panel (a) shows outgoing long wave radiation (OLR) at TOA, and panel (b) shows outgoing SW radiation (OSR) at TOA. Global biases are below %1 of the global time mean; however, local biases are more pronounced. Decrease in OLR and increase in OSR over the Maritime continent is consistent with increase in high cloud cover in the region (not shown). Increase in OLR and decrease in OSR in the subtropical areas off western coasts of continents is consistent with decrease in stratocumulus cloud cover (not shown). These changes in cloud cover are also consistent with increase in downwelling SW at

the surface in the stratocumulus regions and decrease over the Maritime continent, shown on the panel (c) of Figure 2, with global time mean biases being about 0.2% of the global average.



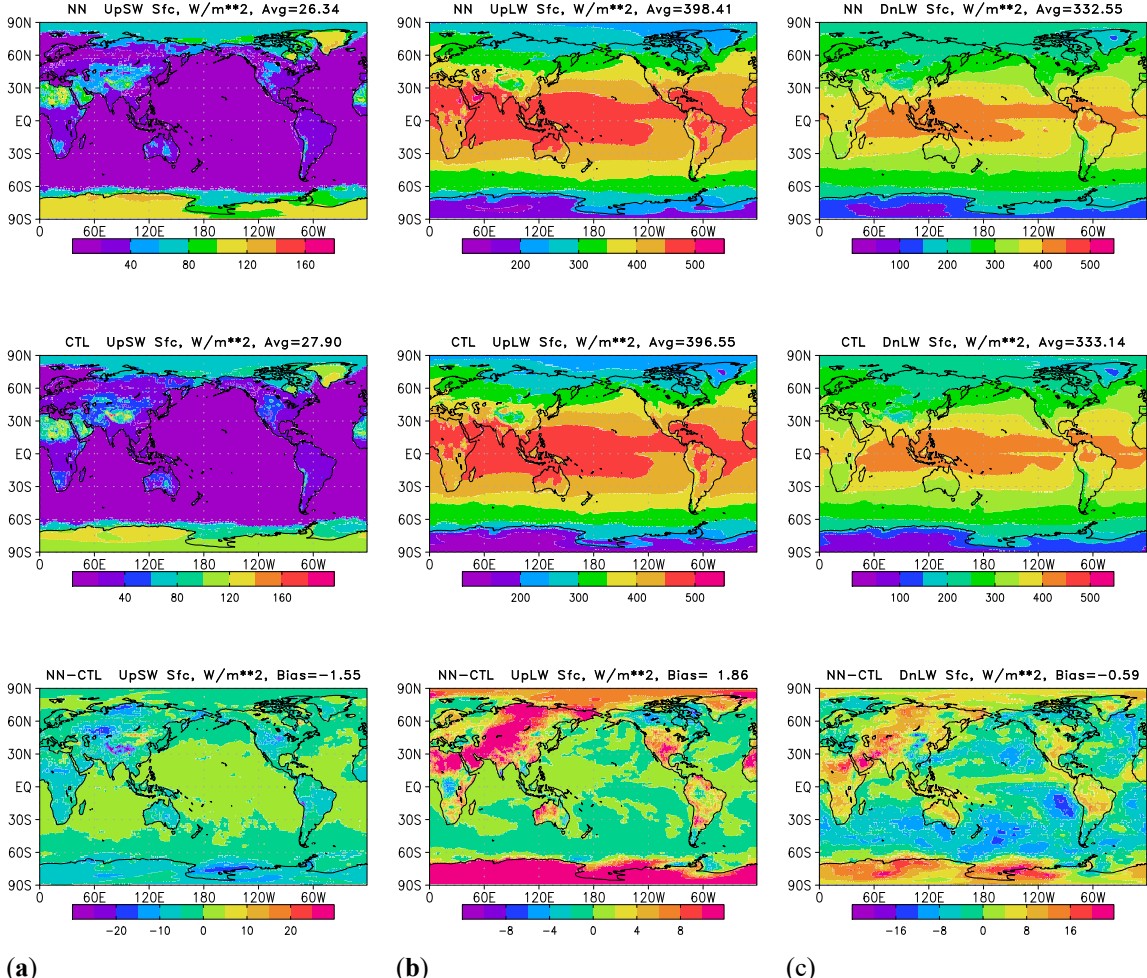

**Figure 3. Zonal and time mean over 12-month AMIP-like run covering 2018 for: (a) Upwelling SW radiation at the surface; (b) Upwelling LW radiation at the surface; (c) Downwelling LW radiation at the surface. Upper row – results produced by HGFS, medium – by GFS, and the lower row the difference (HGFS – GFS). Vertical coordinate shows model level number**

Figure 3(a) shows upwelling SW radiation flux at the surface. Global mean negative bias is almost 5% of the global average value, with negative biases prevalent over continents and extratropical oceans and positive biases over tropical oceans. Upwelling LW at the surface (Figure 3(b)) is biased high by about 0.5% of the global mean value, with positive biases over most of the continents, polar areas and most of tropical oceans, and negative biases in the midlatitude oceans, northern Canada and Alaska, as well as Barents and Norwegian Seas. Downwelling LW at the surface (Figure 3(b)) is biased low globally by approximately 0.5%.





## 4. Discussion

Developing a stable and robust ML/NN application is a multifaceted problem that requires deep understanding of multiple technical aspects of the training process and details of NN architecture. Many techniques for stabilization of hybrid statistical-
deterministic models have been developed. Compound parameterization has been proposed for climate and weather modeling applications where an additional NN is trained to predict errors of the NN emulator, and, if the predicted error is above a certain threshold, compound parameterization falls back to calling the original physically-based scheme (Krasnopolsky et al, 2008). Stability theory was used to identify the causes and conditions for instabilities in ML parameterizations of moist convection when coupled to idealized linear model of atmospheric dynamics (Brenowitz et al, 2020). An NN optimization via
random search over hyperparameter space resulted in considerable improvements in stability of subgrid physics emulators in the Super-parameterized Community Atmospheric Model version 3.0 (Ott et al, 2020). A coupled online learning approach was proposed where a high-resolution simulation is nudged to the output of a parallel lower-resolution hybrid model run, and the ML-component of the latter is retrained to emulate tendencies of the former, helping to eliminate biases and unstable feedback loops (Rasp, 2020). Random forests approach was successfully used to build a stable ML parameterization of
convection (Yuval and O'Gorman, 2020). Physical constraints were used to achieve stability of hybrid models (e.g., Yuval et al, 2021; Kashinath et al, 2021). There are several basic issues that should be pointed out in this respect and that are important to discuss here.

### 4.1    Reusing ML methodologies from different fields

From the mathematical point of view, model physics and individual parameterizations are mappings

$$Y = M(X); \ X \in \Re^n, \ and \ Y \in \Re^m \tag{1}$$

where $n$ and $m$ are the dimensionalities of the input and output vector spaces correspondingly. Therefore, emulating existing parameterizations and developing new ones using ML techniques is a mapping approximation problem.

Reusing methodologies from one ML field in another is often suggested as a particularly powerful tool (e.g., Boukabara et al,
2019), allowing to leverage existing knowledge developed in other ML applications. One example of such approach is transfer learning. However, this powerful tool should be used carefully. ML has been applied to many different problems such as image processing, classification, pattern recognition, asynchronous signal processing, feature detection, data fusion, merging, morphing, mapping, etc. From the mathematical standpoint, these problems belong to different mathematical classes. Applying a methodology developed for a problem belonging to a mathematical class different than the class of the problem at hand may
not necessarily be justified. Therefore, caution should be exercised when applying techniques designed for problems other than mapping approximation to development of model physics emulators or new ML-based parameterizations.



## 4.2    Shallow vs. deep neural networks: complexity and nonlinearity

Application of shallow NNs to the problem of mapping approximation has thorough theoretical support. The universal approximation theorem proves that an SNN is a generic and universal tool for approximating any continuous and almost
continuous mappings under very broad assumptions and for a wide class of activation functions (e.g., Hornik et al, 1990; Hornik, 1991). We are not aware of similarly broad results for deep NNs (DNNs), however specific combinations of DNN architectures and activation functions have theoretical support (e.g., Leshno et al, 1993; Lu et al, 2017; Elbrachter et al 2020). Until there is a universal theory, it has been suggested to consider DNN a heuristic approach since, in general, "from the theoretical point of view, deep network cannot guarantee a solution of any selection problem that constitute complete learning
problem" (Vapnik, 2019). These considerations are important to keep in mind when selecting NN architecture for the emulation of model physics or its components

Next, we compare some properties of DNNs and SNNs to further emphasize their differences and to point out some properties of DNNs that may lead to instabilities in deterministic models coupled to DNN-based model components.

To avoid overfitting and instability, complexity and nonlinearity of approximating/emulating NN should not exceed complexity and nonlinearity of the mapping to be approximated. A measure of the SNN complexity can be written as (see below for explanation),

$$\mathbb{C}_{SNN} = k \cdot (n + m + 1) + m \qquad (2)$$

where $n$ and $m$ are the numbers of the SNN inputs and outputs, and $k$ is the number of neurons in a single hidden layer. The complexity of the SNN (Equation 2) increases linearly with the number of neurons in the hidden layer, $k$. For given numbers of inputs and outputs there is only one SNN architecture/configuration with a specified complexity $\mathbb{C}_{SNN}$.

For the DNN complexity, a similar measure of complexity can be written as (again, see below for explanation),
$$\mathbb{C}_{DNN} = \sum_{i=0}^{K} k_{i+1}(k_i + 1) \qquad (3)$$

where $k_i$ is the number of neurons in the layer $i$ ($i = 0$ and $i = K$ correspond to the input and output layers, respectively). The complexity of the DNN (Equation 3) increases geometrically with the increasing number of layers, $K$.

Both $\mathbb{C}_{SNN}$ and $\mathbb{C}_{DNN}$ are simply the number of parameters of the NN that are obtained during SNN/DNN training. While there
exists a one to one correspondence between the SNN complexity, $\mathbb{C}_{SNN}$, and the SNN architecture, given the fixed number of neurons in the input and output layers, correspondence between the DNN complexity, $\mathbb{C}_{DNN,}$ and the DNN architecture is multivalued: many different DNN architectures/configurations have the same complexity $\mathbb{C}_{DNN}$ given the same size of input and output layers. Overall, controlling complexity of DNNs is more difficult than controlling complexity of SNN.




For an SNN given by the expression

$$y_j = b_j^1 + \sum_{i=1}^{k} a_{ji}^1 \cdot t_i \ , \ \ j = 1, \cdots, m \ ,$$

where $n$, $m$, and $k$ are the same as in Equation (2), nonlinearity increases arithmetically or linearly with addition of each new hidden neuron, $t_i = \phi(b_i^0 + \sum_{s=1}^{n} a_{is}^0 \cdot x_s)$, to the single hidden layer of the NN.

For a DNN, symbolically written as

$$Y = X^{n+1} = B^n + A^n \cdot \phi\left(B^{n-1} + A^{n-1} \cdot \phi\left(B^{n-2} + A^{n-2} \cdot \phi\left(B^{n-3} + \cdots \phi(B^0 + A^0 \cdot X)\right)\right)\right),$$

each new hidden layer/neuron introduces additional nonlinearity on top of nonlinearities of the previous hidden layers; thus, the nonlinearity of the DNN increases geometrically with addition of new hidden layers, much quicker than the nonlinearity of the SNN. Thus, controlling nonlinearity of DNNs is more difficult than controlling nonlinearity of SNNs. The higher the nonlinearity of the model the more unstable and unpredictable generalization is (especially nonlinear extrapolation that is an ill-posed problem).

DNNs are a very powerful and flexible technique that is extensively used for emulation of model physics and its components (Kasim et al, 2020). Discussion of its limitations can be found in Thompson et al (2020). The arguments listed here are intended to point out possible sources of instability of DNNs in the models and the need for careful handling this very sensitive tool.

**4.3 Continuously vs. not continuously differentiable activation functions**

Universal approximation theorem for SNNs is satisfied for a wide class of bounded, non-linear AFs. Note, that many popular AFs used in DNN applications, e.g. variants of ReLU, do not belong to this class. However, for a specific problem of mapping approximation, it may be useful to consider additional restrictions on AFs.

If the AF is almost continuous, or, in other words, has only finite discontinuities (e.g, step function), the first derivative (Jacobian) of the NN using this AF will be singular. If the AF is not continuously differentiable (e.g, ReLU), its first derivative will not be continuous (will have finite discontinuities), and so will be the NN Jacobian. Using a non-continuously differentiable NN as a model component may lead to instability, especially if the Jacobian of this component is calculated in the model. Using gradient-based optimization algorithms for training such NNs may be challenging due to discontinuities in gradients.

If the AF is monotonic, the error surface associated with a single-layer model is guaranteed to be convex, simplifying the training process (Wu, 2009). When AF approximates identity function near the origin (i.e. $\phi(0) = 0.$, $\phi'(0) = 1$, and $\phi'$ is continuous at 0), the neural network will learn efficiently when its weights are initialized with small random values. When the



activation function does not approximate identity near the origin, special care must be used when initializing the weights (Susillo and Abbott, 2014).

It is noteworthy that the sigmoid and hyperbolic tangent AFs, popular in SNN applications, meet all aforementioned criteria.
Additionally, in the context of emulation of model physics parameterizations, these AFs provide one of the lowest training losses, as compared to other AFs (Chantry et al, 2020).

## 4.4 Preparation of training sets

Specifics of training set design may impact stability of the NN as well. We would like to point out a few aspects of training set preparation that, in our experience, are of relevance to development of ML-based components of geophysical models.


A general rule of thumb when it comes to fitting statistical models to data is that the number of records in the training set should be at least as large as the number of the model parameters, or, in the context of current discussion, as the NN complexity introduced in Section 4.2. As a consequence, NNs of larger complexity require larger training sets to approximate a given mapping. To use DNN as an example, as the complexity of DNN, $\mathbb{C}_{DNN}$ (Equation 3), increases geometrically with the number
of DNN layers, so does the amount of data required for the DNN training (Thompson et al, 2020)

We also find that comprehensiveness of the training set is an important contributing factor to the generalization capability of the NN. In the context of application at hand, comprehensiveness of the training set means that it should encompass as much of complexity of the underlying physical system as permitted by the numerical model that hosts the NN. In practice, it translates
into sampling diurnal, seasonal, and annual variability, as well as states introduced by boundary conditions, e.g greenhouse gas and aerosol concentrations, realistic orography, and surface state. Inclusion of events of special interest, e.g hurricanes, snow storms, droughts, extreme precipitation events etc., is beneficial as well.

Care should be taken of proper sampling of the training data. For example, saving the training data set on a Gaussian longitude-
latitude grid will result in overrepresentation of polar areas, and data must be resampled to get more uniform representation over the globe.

Purging and normalization of inputs and outputs are important. Constant inputs and outputs must be removed: from the standpoint of mapping emulation, constants carry no information about the input-to-output relation; however, with incorrect
normalization, they may become a source of noise during training. Normalization of inputs and outputs strongly affects NN training. More specifically to the present application, if some inputs or outputs of an NN are vertical profiles of a physical variable, as is common in geophysical models, the profiles should be normalized as a whole, as opposed to as a collection of



independent variables, for the NN to better capture correlations and dependencies between the levels of the profile (Krasnopolsky, 2013).

**4.5 Handling stochastic behavior**

Model physics may exhibit stochastic behavior for the following reasons: (1) the parametrized process is fundamentally stochastic, (2) a stochastic technique (e.g., a Monte Carlo method) is used in mathematical formulation of the parametrization, and (3) contribution of subgrid processes, or uncertainties in the data that are used to define the mapping (Krasnopolsky, 2013; Krasnopolsky et al, 2013)

If the stochastic nature of a parameterization is overlooked, it can become the source of "noise" that can significantly reduce the accuracy of the emulating NN and lead to instability of the host model. Ensembles of NNs were proposed as an approach to emulation of stochastic mappings (Krasnopolsky et al, 2013).

**5. Conclusions**

One of the major challenges in development of ML/AI-based parameterizations for multi-dimensional non-linear forward environmental models is ensuring stability of the coupling between deterministic and statistical components. This problem is particularly acute for neural network-based parameterizations since, in theory, generalization to out-of-sample data is not guaranteed, and, in practice, previously unseen inputs lead to unphysical outputs of the NN-based parameterization, often destabilizing the hybrid model even in idealized simulations.

Shallow NN-based emulators of radiative transfer parameterizations developed almost a decade ago for a state-of-the-art GCM are stable with respect to substantial structural and parametric change in the host model: when used in two seven months long experiments with the new model, they not only remain stable, but generate realistic output. Two types of modifications of the host model that NN emulators cannot tolerate are the change of the model vertical resolution, and the change in number of model prognostic variables due to, in both cases, alteration of dimensionality of phase space of the mapping (parameterization) and of the emulating NN. After the changes of this nature are introduced into the host model NN emulators must be retrained.

We conjecture that comprehensiveness and realism of the training data set used during development of AI/ML model components, along with careful control of NN complexity, and a synergistic collaboration between both ML and modeling experts, are important factors contributing to generalization capability of the ML component and stability of the model utilizing it, as well as the ability of the ML component to remain functional without re-training despite substantial changes in the host model.





*Code availability.* The source code of NCEP GFS atmospheric model component using the full physics NN emulator, including
the file with trained NN coefficients, is available in the GitHub repository https://github.com/AlexBelochitski-
NOAA/fv3atm_old_radiation_nn_emulator, and is also archived on Zenodo: doi: 10.5281/zenodo.4663160

*Author contributions.* Conceptualization, A.B. and V.K.; methodology, A.B. and V.K.; data for training, A.B.; NN training
V.K.; NN validation, A.B.; analysis of results A.B. and V.K.; writing A.B. and V.K. All authors have read and agreed to the
published version of the manuscript.

*Competing interests.* The authors declare no competing interests.

*Acknowledgments.* The authors would like to thank Drs. Ruiyu Sun and Jun Wang for valuable help with practical use of NCEP
GFS and for useful discussions and consultations. We also thank Drs. Jack Kain, Fanglin Yang, and Vijay Tallapragada for
their support.

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
