# Peer review of "Robustness of Neural Network Emulations of Radiative Transfer Parameterizations in a State-of-the-Art General Circulation Model"

_Geoscientific Model Development, 2021_

## Author Comment (AC1)

The authors thank the reviewers for the detailed and constructive comments. In the following, the reviewers' comments are in red and our replies are in black.

**Reply to Edoardo Bucchignani**

This paper deals with evaluation of ML emulators of radiative parameterizations in General Circulation Models. In particular, authors aim to verify if the results of model calculations with original parameterization and with its ML emulator are identical. Comparisons have been performed between results obtained with GFS at C96 horizontal resolution and an hybrid deterministic-statistical GFS (HGFS), in which the original parameterization schemes have been replaced with shallow-NN-based emulators of LW and SW RRTMG developed for CFS.

While this assessment is correct, our major intent for this paper is the discussion of stability and robustness of an ML emulator that, in essence, was trained using the data from one state-of-the-art global atmospheric model and successfully introduced in another state-of-the-art GCM and integrated online for a long time. A number of recent publications (referred to in the manuscript) report instability of atmospheric models coupled to the ML emulators (or ML parameterizations) specifically designed for these models by using the host model generated data. This problem presents a major practical limitation for using ML emulators online in numerical climate and weather simulations. In this paper we demonstrate that a shallow NN (SNN) based emulator can be robust and stable enough to be integrated for a long time and with satisfactory accuracy even in a model significantly different from that for which the emulator was designed, and discuss aspects of the emulator design and training process that, in our experience, are contributing factors to the emulator's stability. Given the changes in the emulator's host model, we cannot and do not expect the output of parallel runs to be identical. However, we show that the HGFS runs are stable and produce different but physically realistic results. To make sure that our intended message is conveyed to the reader clearly, we will emphasize the above points in the abstract and in the introduction.

The problem has been explained clearly, the topic discussed in this work is very interesting, so the manuscript is potentially suitable for publication. It furnishes a good contribution in the context of existing literature. However, before I can recommend the publication, several issues must be addressed.

My main concern is about the organization of the paper, since in the present form it is not so easy to be read. In particular:

The Introduction is too long and does not allow the reader to have a clear frame of the problem under study. I suggest to split it into two parts: one about the main concepts, a synthesis of the state of art, and the aims of the present work; another one about all the other technical details of the emulators developed at ECMWF, NCEP et al…

We agree and will split the existing introduction into two parts along the lines of the reviewer's recommendation: Introduction (current lines 1-70, 150-163)  and  Survey on

Technical Aspects of Existing ML Emulators of Radiative Transfer Parameterizations (current lines 71-149).

The title of Section 2 does not reveal the content. It is true that authors explain the differences between the two models, but they also explain how they have modified the setup of GFS in order to perform the tests described.

We agree and will change the title to Design of Numerical Experiments with GFS v16

The discussion section is interesting, but the concepts described are quite generical and not specifically linked to the results obtained in the present work. In particular, concepts expressed in subsection 4.1 would fit better into the Introduction. Subsection 4.2 is mainly theoretical: I suggest to shorten it and to move it in another section.

We agree (with one exception) and will modify Section 4 to sharpen its focus on the design aspects of the ML emulator that, in our experience, contribute to the emulator's stability but, to the best of our knowledge, have not been highlighted in literature before. Following the reviewer's recommendation, we will move the first paragraph of subsection 4.1 to Introduction between current lines 149 and 150 and remove what's left of subsection 4.1 altogether. To address the reviewer's comment that the discussion is not "specifically linked to the results obtained in the present work", we will remove subsection 4.5 since radiative transfer parameterizations used in NCEP's atmospheric models do not exhibit stochastic behavior. We will edit subsection 4.2 to make it less generic and more focused on the practical reasons for why differences between SNNs and DNNs contribute to the stability of the ML emulator.

We think that subsection 4.2 is an important part of the paper for the following reasons. All publications reporting instability of ML emulators coupled to atmospheric models use DNNs. The stable emulator discussed in this study and the stable emulator developed at ECMWF (and used in the previous versions of the operational data assimilation system there) are based on SNNs. Therefore, examination of differences between SNNs and DNNs is pertinent to this study. Moreover, all practical recommendations of Section 4 (remaining after we will introduce changes described in the previous paragraph) were incorporated in the design of our stable and robust emulator. We will modify the beginning of Section 4 to put additional emphasis on these points.

Analysis of results is mainly qualitative and performed only through analysis of Figures. I suggest to add some Tables with numerical values of specific indicators (bias, rmse etc…) in order to have a detailed quantification of the differences. In other words, Results section must be strengthened, since it is quite weak in the present form.

We largely agree. All figures presented in the paper have the lower panel that shows the difference between GFS and HGFS runs. Color bars that are shown below each panel, as well as biases and units shown above each panel, provide quantitative information about these differences. The purpose of the figures is to show that differences are reasonably small and results of HGFS runs are physically meaningful, but we cannot expect the runs to be identical (see our response above). We will add tables with numerical values quantifying the differences between the GFS and HGS runs, as well as will make numbers below the color bars and above the figure better visible.

Similarly, the Conclusion section is too generical and quite weak.

We will modify the Conclusions section to have more emphasis on practical decisions used in the design of the stable and robust ML emulator presented in the paper.

Specific comments

Line 60: What do you mean with "results are physically identical" ?

We will replace "are physically identical" with "are close to each other by a metric appropriate for an application at hand as to be identical for the practical purposes."

Lines 70-71: I suggest to introduce a sentence to make a better link between the concept expressed in the paragraph 65-70 with the one expressed afterwards.

We agree. As a part of re-organization of the current Introduction suggested by the reviewer and outlined above, current lines 71-149 will be moved into a separate section "Survey on Technical Aspects of Existing ML Emulators of Radiative Transfer Parameterizations", eliminating the need to connect them to the paragraph 65-70.

Lines 151-154: In my opinion, this paragraph is not properly linked with the previous text, but rather seems a "stand alone" one.

We agree. Once the current lines 71-149 are moved to a separate section, the need to connect them to the lines 151-154 will be eliminated.

Lines 195-196: Avoid using stand-alone sentences.

We will expand lines 195-196 along the following lines: "Concentrations of radiatively active gases are important inputs to radiative transfer schemes, and, more generally, are important parameters of the Earth system. From the standpoint of ML emulator training, change in these parameters leads to a change in phase space of the host model, potentially necessitating retraining of the emulator to ensure its accuracy and stability. $CO_2$ concentration values used during training of 2011 emulators ranged from 350 to 380 ppmv between the years 1990 and 2006, respectively. In our current experiments spanning 2018, $CO_2$ concentration was about 409 ppmv, or about 10% higher on average than in the training set."

Line 211: "e-folding" is scientific jargon. I suggest to avoid it.

We will change "relax to climatology with 90-day e-folding time scale" to "exponentially relax to climatology on a 90-day time scale"

Lines 203-208: "In addition … or HGFS". This paragraph does not fit well into the "Results" section. I would move it into the previous section.

We agree. This paragraph will be incorporated into the previous section.

Line 207: "with radiative transfer emulators developed in 2011". This sentence is too generical, please use more precise terms.

We will replace ""radiative transfer emulators developed in 2011" with ""radiative transfer emulators developed in Krasnopolsky et al (2008,2010, 2012)"

Line 216-218: "Note, that… experiments". I think that this information is not so relevant for the reader, being more related to management policies of the supercomputing center.

We agree, this sentence will be removed.

Figures 1-2-3: Put a proper label on horizontal and vertical axes. Put the unit of measure on the colorbar.

Will do.

Line 230: Explain the meaning of TOA (Top of Atmosphere).

Will do.

Figures 2-3 (caption): "Vertical coordinate shows model level number." It seems to me that latitude values are shown on the vertical axis, please clarify.

The reviewer is correct, the statement "Vertical coordinate shows model level number" will be removed.

Line 279. Acronym SNN is introduced here for the first time. Explain the meaning.

Will do.

**Reply to Anonymous Reviewer**

Unfortunately, the paper does not convince me in the current form. The paper is separated into two parts. The first part shows that a neural networks emulator that was developed for one model can be used within a different model and obtain reasonable results. This is interesting in principle. However, the new model is changed significantly to allow for the application of the old emulator (e.g. in terms of vertical resolution) and the quality of the solution with the neural network is not evaluated sufficiently.

We address most of the issues raised in this paragraph in our replies in the Major Comments section, however, here we would like to comment on the reviewer's correct observation that "the new model is changed significantly to allow for the application of the old emulator (e.g. in terms of vertical resolution)" . While this statement is true, it is also true that the model configuration used in this paper is also changed very significantly, perhaps much more so, from the 2011 version of NCEP CFS used for generation of data for the training set for the ML emulators discussed in this paper (see the list of model changes from

2011 to 2021 here: https://www.emc.ncep.noaa.gov/gmb/STATS/html/model_changes.html. This document is also referenced in the manuscript) A number of recent publications (referenced in the manuscript) report quickly developing instability of atmospheric models coupled to the ML-based model components that were trained specifically for these models. In this paper we demonstrate that an ML emulator can be stable and produce realistic output when integrated for a long time even after substantial structural and parametric changes to the host model. This stability and robustness of the ML emulator, as well as the design decisions that distinguish it from others, are the focus of this paper. In other words, given the current state of the field and the number and type of changes to the host model, the stability of this emulator is a noteworthy result.

The change in vertical resolution is dictated purely by software engineering constraints: the ML emulator expects input vertical profiles of model variables to be given as 64 element arrays, because until March 2021 NCEP global atmospheric models used 64 vertical levels.

The second part is a long discussion on what should be done to obtain emulators that are working in different environments. While the discussion is interesting in principle, it is not backed up by the results of the first part (e.g. as only a single (shallow) neural network has been tested). The arguments of the discussion are intuitive, but most of them are not backed up by powerful references or by numerical experiments. I therefore think that the paper is not suitable for GMD in the current form. I hope the authors can improve the paper along the following suggestions.

We address this in the next session.

Major comments:

The figures 1-3 do not leave me convinced that the neural network configuration is good enough. Can you show results that would convince me that the GFS version with 64 vertical levels is producing something useful (e.g., by comparing it against the default model with 127 vertical levels)?

Operational global atmospheric models used for weather forecasting at NCEP used 64 levels in the vertical from 2002 to 2021 (see the link in the previous section) before switching to 127 levels in March of this year. Model configuration used in this paper is directly comparable to the current generation of climate models in terms of spatio-temporal resolution and sophistication of model physics and dynamics. In contrast, most publications describing online coupling of atmospheric models to ML components use idealized models, including those publications that report instabilities.

Can you show more diagnostics that compare GFS and HGFS for quantities that are no mean fields or at least provide plots of variability? Can you show vertical profiles that compare the neural network emulator and the default radiation scheme to visualise how different the solutions are?

We will include plots showing temporal variability of model variables. However, we would like to emphasize that assessing the accuracy of the ML emulator is not the goal of this paper, but rather investigation of the fact the emulator remains stable in a long integration

while producing output that looks like planet Earth! Given the extent of structural and parametric changes in the host model, we cannot expect the emulator to be useful for practical applications. In the context of the current state of the field of coupling of atmospheric models to ML components, the stability of this emulator in itself is a noteworthy result.

Each sub-section of section 4 should show a clear connection to the numerical simulations. If this is not possible, the content should be either removed or backed up with a theoretical study.

We will address this issue separately for each subsection in the text that follows.

Section 4.1 should become part of the introduction.

We agree. We will move the first paragraph of subsection 4.1 to Introduction between current lines 149 and 150 and remove what's left of subsection 4.1 altogether.

Section 4.2 should be backed up by experiments because it otherwise remains speculative.

We disagree. Statements of subsection 4.2 follow directly from the analytical forms of both shallow and deep NNs. All publications reporting instability of ML emulators coupled to atmospheric models use DNNs. The stable emulator discussed in this study and the stable emulator developed at ECMWF (and used in the previous versions of the operational data assimilation system there) are based on SNNs. Therefore, examination of differences between SNNs and DNNs is pertinent to this study. We will edit subsection 4.2 to make it less generic and more focussed on the practical reasons for why differences between SNNs and DNNs contribute to the stability of the ML emulator.

The argument that SNNs are better than DNNs as they are more generalisable is not backed up with simulations

Respectfully, we do not make such an argument.

and we also do not learn how non-linear the ML solution needs to be from the paper.

While this statement is correct, estimation of non-linearity of a mapping given by a set of input-output pairs is outside of the scope of this paper.

Section 4.3 is also not backed up by any experiments.

Statements of subsection 4.3 are either backed by references given therein or follow from the analytical form of activation functions discussed in this subsection.

Section 4.4 and section 4.5 read like chapters of a textbook and are not very relevant for a paper that is not training new emulators.

We partially agree. We will remove subsection 4.5 because radiative transfer parameterizations used in NCEP's atmospheric models do not exhibit stochastic behavior. However, subsection 4.4 describes major practical decisions that we made while designing our stable emulators.

Minor comments:

The description of radiation schemes in general is rather long with machine learning only starting at l.55

We think that contextualizing the ML emulation approach among other numerical and, especially, statistical techniques used for acceleration of atmospheric radiative transfer calculations is appropriate for a journal with an emphasis on geoscientific model development as opposed to the journal with an emphasis on machine learning.

The neural network solutions that are used in the paper are not described in sufficient detail in Section 2.

While it is true that the description of neural network based emulators used in the paper is abbreviated, the lines 86-120 contain references to publications that document the emulators in detail (Krasnopolsky et al, 2008,2010, 2012).

l.65: I do not think that this very general statement holds. I am sure that I could break forecast skill with an unbiased, random, uncorrelated error if I wanted to.

We agree that this statement will not hold for perturbations of arbitrary magnitude. Therefore, we will quantify this statement in the following manner: "Unbiased, random, uncorrelated errors in radiative heating rates with magnitudes as large as the net cooling rate do not statistically affect forecast skill of an atmospheric model" See Section 4 of Pincus and Stevens (2013, JAMES) for the discussion of the amplitude of unbiased noise applied to the radiative heating rates.

l.77: I do not understand "blackbody cloud"

We will replace "black body cloud" with a "cloud with the emissivity of unity".

l.118: Why do you need to store the entire training data set for regression trees?

Because a regression tree selects an answer from this set on each application.

Figure 1: I assume that this does not cover the entire year 2018 but rather 1st Feb 2018 – 1st Feb 2019, correct?

Yes, this is correct, we will modify the text accordingly.

Figure 2 and 3 are not "zonal means" as indicated in the captions. The vertical coordinate is not the "model level number".

The reviewer is correct, the statement "Vertical coordinate shows model level number" and references to zonal mean will be removed.

l.208: " hybrid deterministic-statistical GFS" -- Why do you call it "hybrid"?

The term "hybrid model" refers to a physics-based numerical model where some of the parameterization schemes are ML based.

l.248: This should be moved to the introduction.

We agree. We will move lines 248-262 to introduction and instead point out here that the goal of Section 4 is to emphasize the design aspects of the ML emulator that, in our experience, contribute to the emulator's stability but, to the best of our knowledge, have not been highlighted in literature before.

l.185: Yes, OK, the number of possible model states is increased. But how do you know that this is leading to difficulties in the emulation? I do not think that this can be shown mathematically.

Not just the number of possible model states increases, but, more importantly, the shape of the model phase space that the model states occupy changes. In addition, dimensionality of the model phase space almost doubles when Zhao-Carr microphysics is replaced with the GFDL scheme. Projection of the new phase space on the hyper-space with dimensionality of the old phase space will almost certainly not coincide with the old phase space, resulting in input vectors sampled from this projection being outside of the domain of the emulator's original training data set.

Interestingly, we started out with a model configuration that used GFDL microphysics along with our ML emulator, only to find unphysical values in some of the emulator's outputs. For example, the flux of outgoing longwave radiation at the top of the atmosphere would end up being zero at some locations on the globe. This problem was eliminated by switching back to the Zhao-Carr scheme.

l.281: It is hard for me to believe that there is no universal theorem for DNNs

We will replace "[w]e are not aware of similarly broad results for deep NNs' ' with "[s]imilarly broad results for deep NNs do not exist as of yet (Vapnik, 2019)". See the section "Belief in learning using deep neural network" in the paper of Vapnik (2019) (the reference is in the manuscript). While neither of us is an expert on theoretical foundations of ML, we put trust in this reference given Vladimir Vapnik's reputation.